# Impact of cancer-associated mutations in Hsh155/SF3b1 HEAT repeats 9-12 on pre-mRNA splicing in *Saccharomyces cerevisiae*

Harpreet Kaur[1], Brent Groubert[1], Joshua C. Paulson[1], Sarah McMillan[1,2], Aaron A. Hoskins[1,2]*

1 Department of Biochemistry, University of Wisconsin-Madison, Madison, Wisconsin, United States of America, 2 Integrated Program in Biochemistry, University of Wisconsin-Madison, Madison, Wisconsin, United States of America

* ahoskins@wisc.edu

**Data Availability Statement:** All relevant data are within the manuscript and its Supporting Information files.

## Abstract

Mutations in the splicing machinery have been implicated in a number of human diseases. Most notably, the U2 small nuclear ribonucleoprotein (snRNP) component SF3b1 has been found to be frequently mutated in blood cancers such as myelodysplastic syndromes (MDS). SF3b1 is a highly conserved HEAT repeat (HR)-containing protein and most of these blood cancer mutations cluster in a hot spot located in HR4-8. Recently, a second mutational hotspot has been identified in SF3b1 located in HR9-12 and is associated with acute myeloid leukemias, bladder urothelial carcinomas, and uterine corpus endometrial carcinomas. The consequences of these mutations on SF3b1 functions during splicing have not yet been tested. We incorporated the corresponding mutations into the yeast homolog of SF3b1 and tested their impact on splicing. We find that all of these HR9-12 mutations can support splicing in yeast, and this suggests that none of them are loss of function alleles in humans. The Hsh155[V502F] mutation alters splicing of several pre-mRNA reporters containing weak branch sites as well as a genetic interaction with Prp2 and physical interactions with Prp5 and Prp3. The ability of a single allele of Hsh155 to perturb interactions with multiple factors functioning at different stages of the splicing reaction suggests that some SF3b1-mutant disease phenotypes may have a complex origin on the spliceosome.

## Introduction

RNA splicing is a fundamental process in eukaryotic gene expression. During splicing, intron regions of precursor mRNAs (pre-mRNAs) are identified precisely and are excised with concurrent ligation of flanking exons to form mature mRNAs. Splicing is catalyzed by an extraordinarily complex and highly dynamic ribonucleoprotein (RNP) machine, the spliceosome [1]. The spliceosome is composed of five U-rich small nuclear ribonucleoproteins (the U1, U2, U4, U5, and U6 snRNPs). Each snRNP contains a short U-rich small nuclear RNA (snRNA), 7 Sm or like Sm (Lsm) proteins, and several other proteins specific to each snRNP. The splicing reaction itself is a two-step transesterification process: the first step involves cleavage of the 5'

**Funding:** This work was supported by the National Institutes of Health (R01 GM112735 to AAH; www.nigms.nih.gov) and the Edward P. Evans Foundation (EvansMDS Discovery Research Grant to AAH; www.evansmds.org). The funders had no role in study design, data collection and analysis, decision to publish, or preparation of the manuscript.

**Competing interests:** The authors have declared that no competing interests exist.

splice site (5' SS) of the intron by nucleophilic attack of the branch site (BS) adenosine [1]. During the second step, the 3' hydroxyl group of the cleaved 5' exon carries out a nucleophilic attack at the 3' splice site (3' SS), producing ligated exons and a free lariat intron. Alteration in recognition of the intron/exon junctions may result in splicing dysregulation [2]. Several human diseases, including many different cancers, have been linked to changes in alternative splicing regulation [3, 4]. Recent studies have also found that genes encoding splicing factors are frequently mutated in a number of cancers including chronic lymphocytic leukemia (CLL), myelodysplastic syndromes (MDS) and uveal melanoma [5].

SF3b1, the largest subunit of SF3b subcomplex of the U2 snRNP, is known to play a primary role in intron recognition during spliceosome assembly [6–11]. The N-terminal domain (NTD) of human SF3b1 contains regions which interact with the U2AF, Tat-SF1, and SF3b14 splicing factors (**Fig 1A**) [6, 11–13]. The C-terminal domain of SF3b1 is composed of 20 HEAT domain repeats (HR) [6, 11]. SF3B1 is highly conserved between humans and the yeast *S. cerevisiae* in which the homologous protein is called Hsh155. As part of the U2 snRNP, SF3b1 associates with the BS region of the substrate pre-mRNA during assembly of the A complex spliceosome [11]. In the A complex, the U2 snRNA base pairs with the BS and the U2 snRNA/BS duplex is clamped in a vise formed by the N- and C-terminal HR of SF3b1 (**Fig 1B**) [14, 15]. The interactions between the SF3b1, SF3Bb14, and U2AF65 near the BS indicate that these three proteins all play roles in identification of the BS. Aberrant BS recognition by U2 snRNP may lead to production and accumulation of altered spliced products and cancer. Additionally, SF3b1 is the molecular target of anticancer drugs such as spliceostatin A [16], pladienolide B [17], and herboxidiene [18], which has further raised interest in understanding SF3b1's roles in splicing.

Recent experiments have shown that MDS-related mutations in SF3b1 promote usage of alternative BS within introns. These alternative BS in turn activate nearby cryptic 3' SS to produce aberrantly spliced mRNAs [19, 20]. MDS mutations cluster in SF3b1 HR 4–7 and overlap with a region of the protein proposed to directly interact with the spliceosomal ATPase Prp5 (HEAT repeats 1–12; **Fig 1A**) [9, 21]. MDS mutations in HR 4–7 alter this interaction when monitored with a yeast two-hybrid assay or by pull-down. Multiple mechanisms have been proposed to describe how MDS mutations in SF3b1 ultimately lead to selection of alternative BS. These mutations may influence splicing by altering interactions with Prp5 or the human splicing factor SUGP1 [22], disrupting SF3b1 conformational change, or by perturbing the direct interactions between the intronic pre-mRNA and SF3b1 itself [23]. It is possible that several of these mechanisms may work in concert to promote usage of alternative BS.

A meta-analysis of splicing factor genes from 33 different tumor types across >10,000 samples uncovered a second site of frequent mutation in SF3b1 [24]. HR9-12 contain hotspots that are enriched in acute myeloid leukemia (LAML; 3 HR9-12 mutations found in 201 samples), bladder urothelial carcinoma (BLCA; 11 HR9-12 mutations found in 438 samples), and uterine corpus endometrial carcinoma (UCEC; 13 HR9-12 mutations found in 564 samples) cancer cells. While some amino acids in HR9-12 such as E862 were mutated in multiple cancers, others were only observed in a single cohort (*e.g.*, all 3 HR9-12 mutations observed in LAML are L833F). We collectively refer to this as the LAML-BLCA-UCEC (LBU) hotspot (**Fig 1A** and **1B**). HR9-12 have also been identified as a Prp5 binding site [21] and are located near a putative interaction with the tri-snRNP protein Prp3 [30]. Bioinformatic analyses of the splicing phenotypes of LBU cancer cells revealed that LBU mutations conferred a phenotype more similar to wild-type (WT) SF3b1 than do MDS mutations. A detailed RNA-Seq analysis of BLCA tumor cells harboring the most prevalent LBU mutation (E902K) showed only a small number of significantly altered splice junctions; however, these aberrant isoforms showed preferential use of downstream 3'SS rather than the upstream sites observed in MDS [24]. Taken together

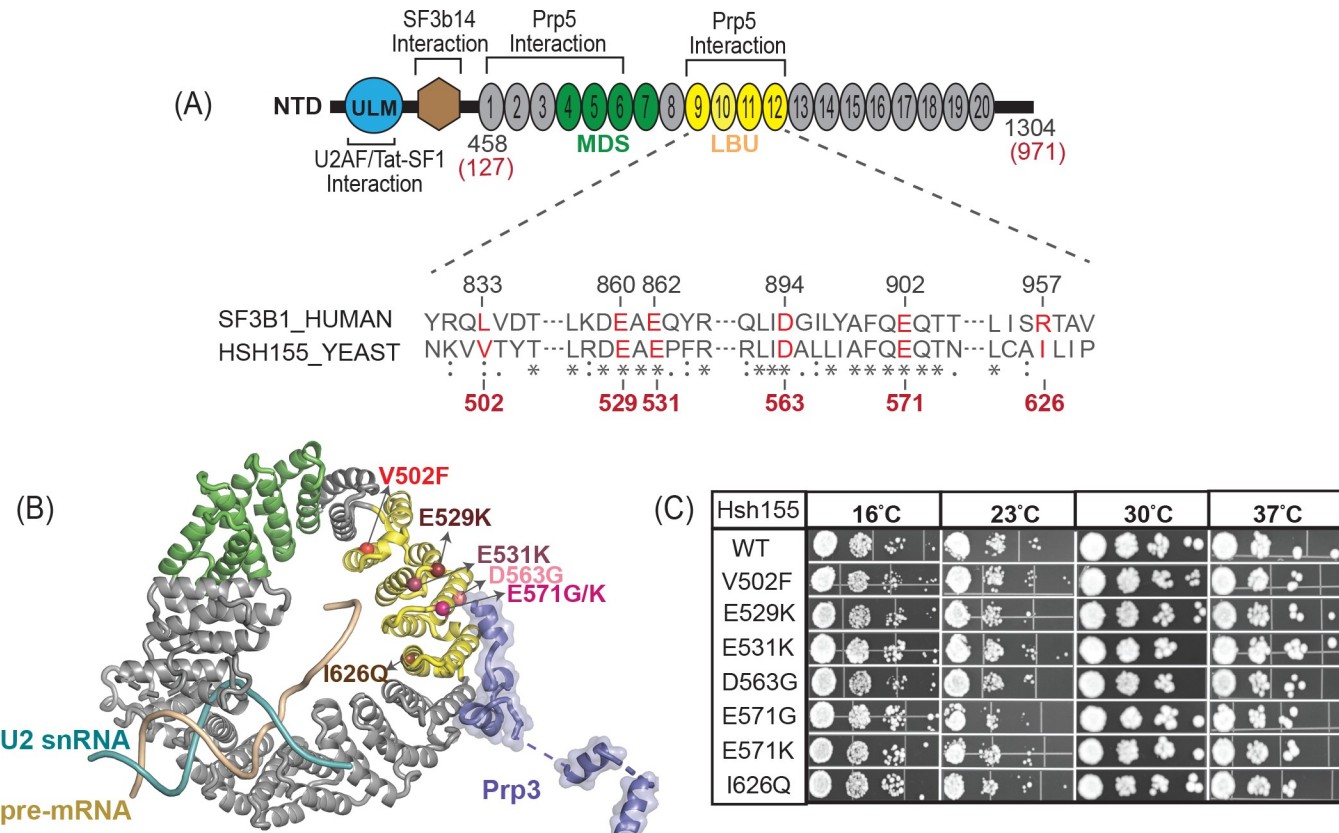

**Fig 1. LBU-mutants of Hsh155 do not affect cellular proliferation.** (A) Schematic of the domain architecture of SF3b1. HR4-7 (green) is a hotspot region for mutations associated with MDS, and LBU-mutants are found in HR9–12 (yellow). The bottom panel shows the alignment of sequences from *H. sapiens* and *S. cerevisiae*. The LBU-mutants studied here are shown in red. The sequence alignment was generated using EMBOSS Needle [40]. (B) SF3b1 LBU mutations mapped to the yeast Hsh155 structure in the B complex spliceosome (PDB:5NRL). Spheres represent the mutations present in HR9-12. Hsh155 HR11-13 are proposed to interact with splicing factor Prp3, which is a component of tri-snRNP. (C) Temperature sensitivity growth assays of Hsh155 LBU-mutant strains plated on YPD. Successive 10-fold dilutions of $OD_{600}$ = 0.5 cultures are shown. Growth assays were carried out in triplicate.

these data suggest that mutations in the MDS and LBU sites may perturb the function of the spliceosome in different ways.

Here, we explore the effect of LBU disease-related mutations in Hsh155 on splicing in yeast. Using a reporter transcript *in vivo*, we show that the hotspot mutation V502F alters splicing of introns containing non-consensus BS in yeast as well as a genetic interaction between the spliceosomal ATPase Prp2 and Hsh155. Other LBU mutants in Hsh155 showed less pronounced phenotypes. Consistent with recent cryo-EM data, we also identify a new yeast two-hybrid interaction between Hsh155 and Prp3 that is disrupted by the V502F mutation. Combined these data indicate that cancer hotspot mutations in Hsh155 have the potential to impact multiple interactions between splicing factors.

## Results

Since the HR domain is highly conserved in between human SF3b1 and yeast Hsh155, we generated mutations in yeast Hsh155 corresponding to six of the most common LBU disease alleles (SF3b1[L833F]/Hsh155[V502F], SF3b1[E860K]/Hsh155[E529K], SF3b1[E862K]/Hsh155[E531K], SF3b1[E902G]/Hsh155[E571G], SF3b1[E902K]/Hsh155[E571K] and SF3b1[R957Q]/Hsh155[I626Q]; Fig 1A). Four of the mutational sites occurred in regions of identical amino acid conservation.

However, L833 in human SF3b1 corresponds to a smaller hydrophobic amino acid in yeast Hsh155 (V502) and R957 in human SF3b1 is not conserved in yeast (I626 at the corresponding position). We also included the mutational site SF3b1$^{D894G}$/Hsh155$^{D563G}$ in this study since it occurs within the HR9-12 region [24]. The SF3b1$^{D894G}$ mutation has been found in skin cutaneous melanoma as well as in CLL patient samples [25]. For simplicity, we refer to the entire mutant series as the "LBU mutants".

We deleted the chromosomal *hsh155* gene in haploid strain of *S. cerevisiae* and maintained cell viability with wild type (WT) Hsh155 expression from a low copy URA3/CEN6-containing plasmid [9]. Next, we transformed the TRP1/CEN6-containing plasmids containing the above mutant alleles into this yeast strain and then carried out 5-FOA selection of the resulting transformants. All transformants were viable after 5-FOA selection, and the genotypes were confirmed by isolating the TRP1/CEN6 plasmids from each resulting strain and DNA sequencing of the *hsh155* gene.

## LBU mutants of Hsh155 support yeast growth

We next tested each LBU-mutant yeast strain for growth defects on solid media. The growth of each strain was assayed at four different temperatures ranging from 16˚C to 37˚C. All of the mutant yeast strains were viable at all temperatures when expressing LBU-mutant Hsh155. Moreover, we did not observe any significant differences in cell proliferation between any of the mutants and the WT control at any temperature (**Fig 1C**). We conclude that the LBU-mutant Hsh155 proteins are functional and do not prevent pre-mRNA splicing in yeast.

## A subset of LBU mutants impact usage of non-consensus branch sites

We have previously shown that MDS-mutant Hsh155 yeast strains containing substitutions in HR4-7 do not significantly perturb yeast proliferation [9]. Nonetheless, the MDS-mutants impact splicing of introns containing non-consensus BS [9, 21]. These splicing changes were most frequently observed when the non-consensus BS introduced mismatches in the U2/BS duplex at positions flanking the BS. Since little change was observed if the BS itself was altered, this suggests that mutations alter how Hsh155 interacts with the U2/BS duplex rather than the non-paired and extra-helical BS nucleotide. To test if the LBU-mutants also impact splicing despite the lack of a growth phenotype, we used the ACT1-CUP1 reporter transcript to assay splicing *in vivo*. Splicing of the ACT1-CUP1 pre-mRNA allows for expression of the CUP1 protein and confers Cu$^{2+}$-resistance to sensitized yeast strains (**Fig 2A**) [26]. The degree of Cu$^{2+}$-resistance is directly related to the extent of ACT1-CUP1 pre-mRNA splicing and mRNA formation [26].

We did not observe any differences in Cu$^{2+}$ sensitivity between any of the LBU-mutant strains or WT yeast when an ACT1-CUP1 reporter was used that contained consensus SS, as shown in **Fig 2B**. Nor did we observe any changes in Cu$^{2+}$ resistance when the branch point adenosine was substituted in the reporter (A259G; **Fig 2C**). However, in the presence of ACT1-CUP1 reporters harboring a non-consensus, weak BS (A258U) both the Hsh155$^{V502F}$ and Hsh155$^{D563G}$ mutants showed lower Cu$^{2+}$ resistance (**Fig 2D**). Hsh155$^{V502F}$ additionally showed lower Cu$^{2+}$ resistance with the non-consensus U257C ACT1-CUP1 reporter (**Fig 2E**).

To correlate the observed growth defects with splicing of the ACT1-CUP1 reporter pre-mRNA, we quantified reporter RNA levels by primer extension [9]. We performed primer extension using RNA isolated from strains expressing the WT or Hsh155$^{V502F}$ Hsh155$^{D563G}$ LBU-mutants in the presence of WT, A258U, or U257C ACT1-CUP1 reporters. We isolated the total RNA from each strain and quantified the relative amount of spliced mRNA. The primer extension assays confirm results from the growth assays with the Hsh155$^{V502F}$ and

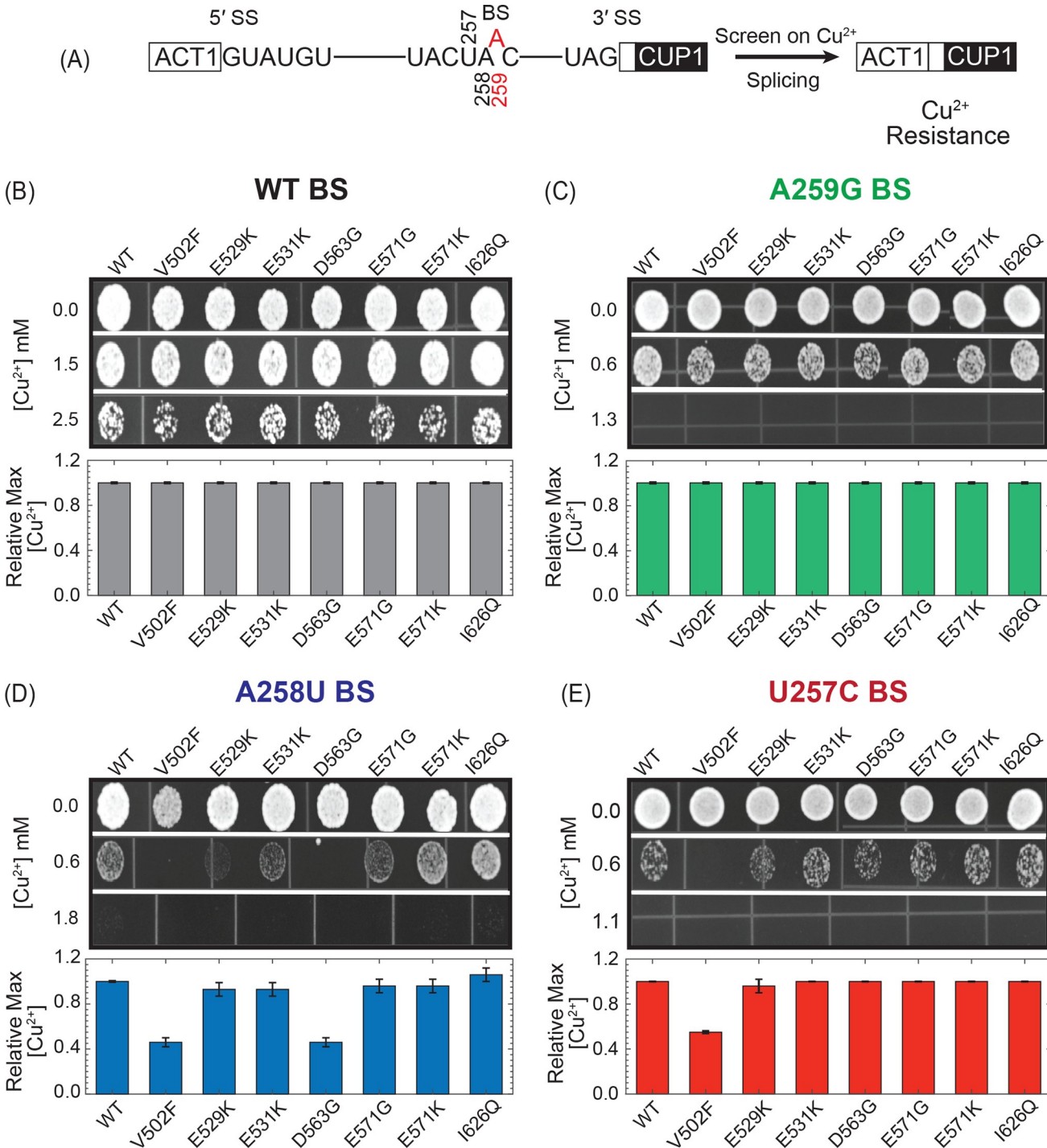

**Fig 2. Impact of LBU-mutants on yeast growth using the ACT1-CUP1 reporter.** (A) Schematic of the ACT1-CUP1 reporter assay and sequences of the yeast 5' SS, BS, and 3' SS. The branchpoint adenosine A259 is colored red. (B) Results of ACT1-CUP1 $Cu^{2+}$ growth assays of Hsh155 WT and mutant strains carrying ACT1-CUP1 reporter plasmids with consensus SS. (C-E) Results of ACT1-CUP1 $Cu^{2+}$ growth assay of Hsh155 WT and mutant strains carrying ACT1-CUP1 reporter plasmids with nonconsensus BS. For panels B-E, representative images of yeast grown on $Cu^{2+}$-containing plates are shown at the top, and the bar graph in the bottom panel represents the average and standard deviation from three replicates of the maximum [$Cu^{2+}$] at which yeast strain survived relative to Hsh155$^{WT}$.

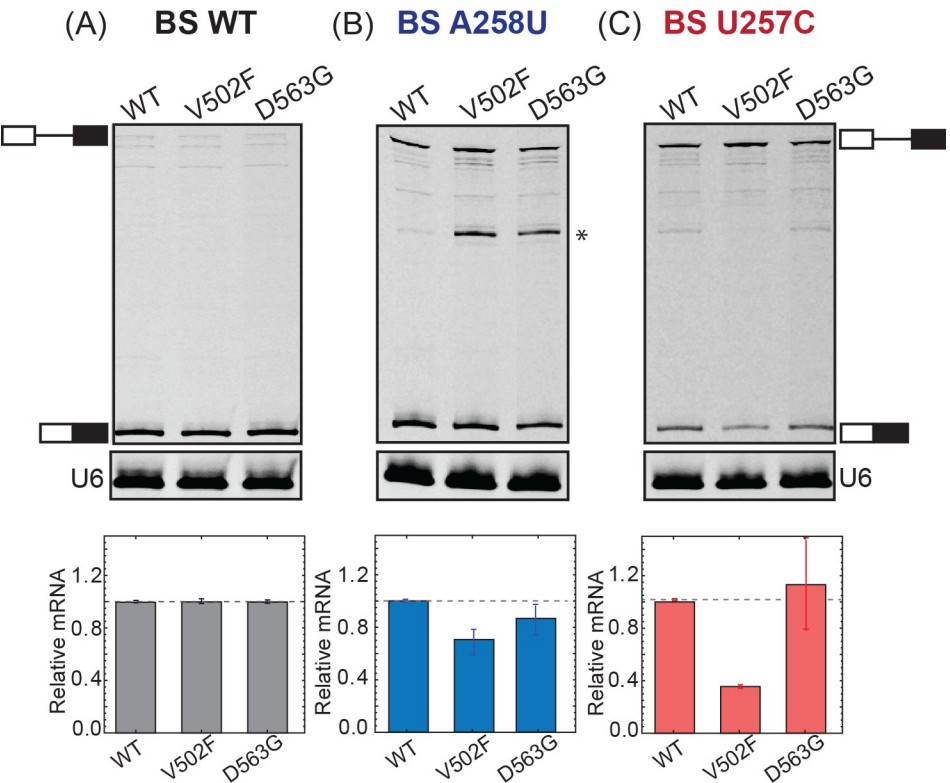

**Fig 3. Primer extension analysis of ACT1-CUP1 splicing.** (A-C) Primer extension analysis of Hsh155 WT and LBU-mutant strains carrying ACT1-CUP1 reporter plasmids. Top panel, PAGE analysis of the primer extension products. The pre-mRNA and mRNA bands are marked. Middle panel, primer extension products from U6 snRNAs analyzed on the same gel as in the top panel and used as an internal control. Bottom panel, quantification of the primer extension results. Each bar graph represents the average and standard deviation of the fraction mRNA [mRNA/(mRNA+pre-mRNA)] observed from three replicates. The asterisk represents a band amplified in the primer extension reaction but not identified. The original image for the gel is provided in S1 Raw images.

Hsh155$^{D563G}$ mutants showing no detectable defect in splicing of the WT RNA (**Fig 3A**). The Hsh155$^{V502F}$ LBU-mutant strain was defective in splicing the A258U and U257C reporters, while only a small decrease in splicing of the A258U reporter was observed with the Hsh155$^{D563G}$ mutant (**Fig 3B and 3C**). Together our results indicate that the majority of the LBU-mutants tested here have little detectable impact on yeast growth and retain splicing activity similar to WT. Only a subset of the mutants (Hsh155$^{V502F}$ and Hsh155$^{D563G}$) show changes in splicing of reporter pre-mRNAs harboring non-consensus BS.

## The effects of multiple HEAT repeat mutations are additive

When multiple MDS-mutant alleles in HR4-7 are combined, their impact on splicing is additive [9]. We wondered if this would also be true for the Hsh155$^{V502F}$ mutant in HR9. We combined the V502F mutation with two previously characterized MDS-mutant alleles: H331D and D450G. These two MDS alleles have opposing effects on splicing of pre-mRNAs with non-consensus BS: H331D inhibits while D450G promotes their splicing. When combined with the LBU-mutant V502F, we observed independent, additive effects. The H331D or D450G mutations still decreased or increased Cu$^{2+}$ tolerance; however, the addition of the V502F mutation lowered the degree of Cu$^{2+}$ tolerance overall.

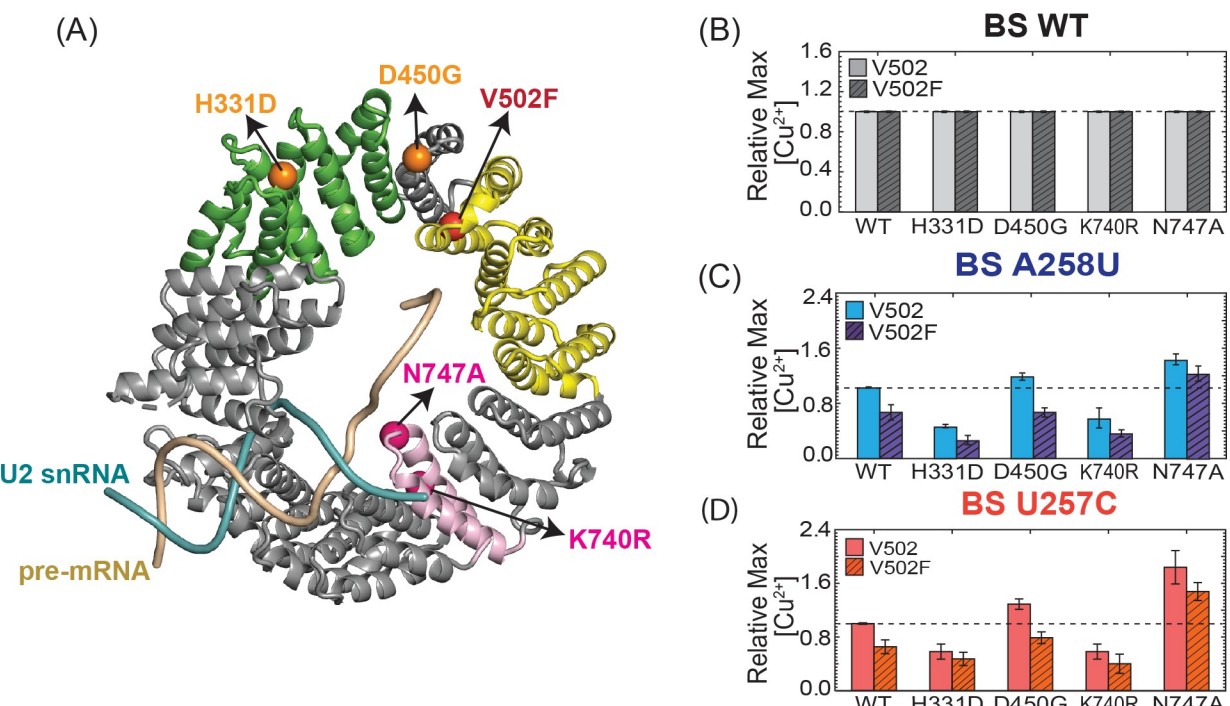

**Fig 4. Effects of Hsh155^V502F in combination with mutations in other HR.** (A) Hsh155 mutations found in different HR are mapped on to the structure of yeast Hsh155 present in the B complex spliceosome (PDB ID: 5NRL). (B-D) Results of ACT1-CUP1 $Cu^{2+}$ growth assays of Hsh155 WT and HR-mutant strains carrying WT or nonconsensus ACT1-CUP1 reporter plasmids. Each bar graph represents the average and standard deviation from three replicates of the maximum $[Cu^{2+}]$ at which yeast strain survived relative to WT Hsh155.

We also tested additional mutations in the RNA duplex binding site of Hsh155. The K740R and N747A mutations in HR15 also decrease or increase $Cu^{2+}$ tolerance in yeast (respectively) when using non-consensus BS reporters [9]. However, these mutations may exert their effects through direct interaction with the U2/BS duplex as opposed to MDS mutation site. The K740R and N747A mutations in HR15 have not been previously studied in combination with those located in distal HR. Therefore, we combined the N747A or K740R mutations in HR15 with the V502F mutation in HR9. The results were similar those observed with the H331D and D450G MDS-mutants in HR5 and HR8 (Fig 4B–4D). The addition of the V502F mutation lowers the degree of $Cu^{2+}$ tolerance in an additive manner in combination with effects from HR15 mutation. From these data, we conclude that the incorporation of multiple mutations in Hsh155 across different HR can result in independent and additive effects on splicing.

## Genetic interaction between the HR9 V502F mutant and the Prp2 ATPase

We previously identified a genetic interaction between some MDS-mutant alleles of Hsh155 located in HR5 and 8 and the spliceosome ATPase Prp2. Prp2 is responsible for release of Hsh155 and associated U2 proteins from the spliceosome during activation (transition of the B^act to B* spliceosome complex) and associates near HR7-8 (Fig 5A) [14, 15, 27]. We tested LBU-mutant Hsh155-expressing yeast strains for genetic interactions with a cold-sensitive (*cs*) allele of Prp2, Prp2^Q548N [28]. We observed no differences in growth for the LBU-mutant yeast strains in the presence of Prp2^Q548N at permissive temperatures (30˚C; Fig 5B). However, Hsh155^V502F exacerbated the *cs* phenotype of Prp2^Q548N. At 23˚C, yeast strains containing both Hsh155^V502F and Prp2^Q548N exhibited a severe defect in growth (Fig 5B). Other LBU-mutants showed little to no change in growth at either 23˚C or 16˚C (not shown) in the

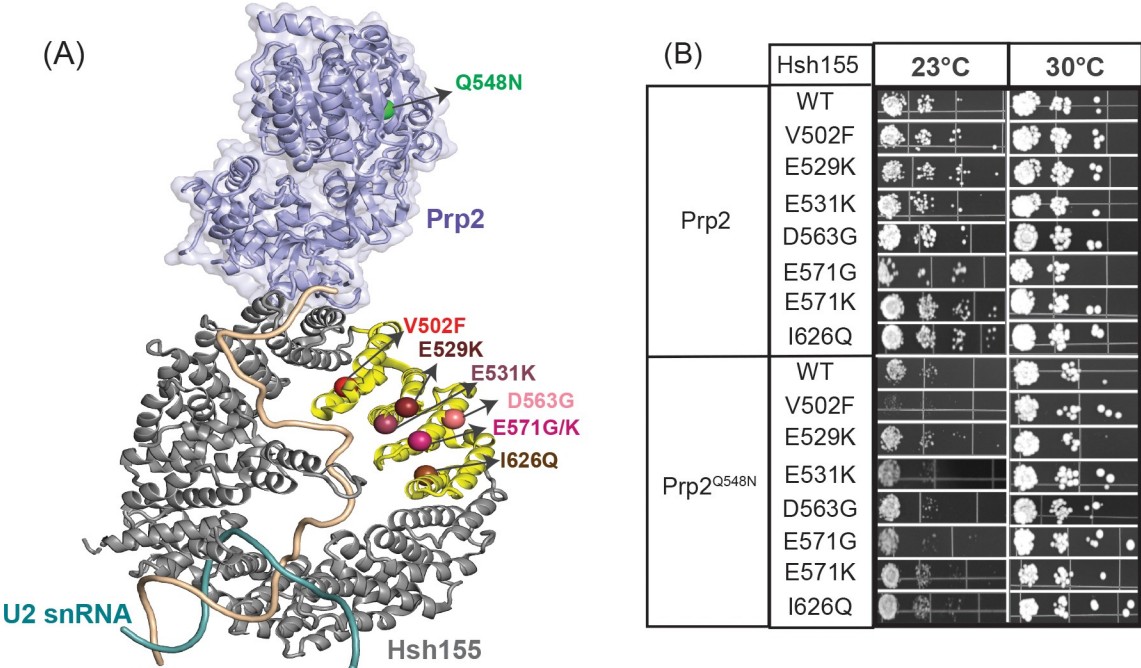

**Fig 5. Genetic interaction between Hsh155$^{V502F}$ and Prp2$^{Q548N}$.** (A) Relative orientation of the Prp2 ATPase (purple) to Hsh155 in the yeast B$^{act}$ spliceosome complex (PDB ID: 5GM6). The site of the Prp2 *cs* mutation Q548N is colored green. Spheres represent the LBU-mutants found in HR9-12 (yellow). (B) Temperature sensitivity growth assays of Hsh155 mutant stains in the presence of WT Prp2 or Prp2$^{Q548N}$ at 23 and 30˚C. Successive 10-fold dilutions of OD$_{600}$ = 0.5 culture are shown, and the assay was carried out in triplicate.

presence of Prp2$^{Q548N}$. This suggests a role for HR9 in Prp2-mediated steps in spliceosome activation in addition to influencing splicing of pre-mRNAs containing non-consensus BS.

## The V502F mutation disrupts Hsh155 interactions with Prp5 and Prp3

The splicing phenotype displayed by the Hsh155$^{V502F}$ mutant could also arise due to changes in interactions with splicing factors in addition to Prp2. In particular, we and others have shown that Hsh155 interacts with the DEAD-box ATPase Prp5 [9, 21]. Prp5 is essential for formation of the U2/BS RNA duplex [19] and mutations in Prp5 can alter BS usage. Tang and coworkers recently defined the Prp5-interacting region of Hsh155 to include HR1-6 and HR9-12 which contains the site of the V502F substitution [21]. To test the impact of Hsh155$^{V502F}$, we carried out a yeast two-hybrid assay for Hsh155/Prp5 interactions as previously described [9]. WT Hsh155 and Prp5 readily form a two-hybrid interaction. This interaction is not observed when Hsh155$^{V502F}$ is used (**Fig 6A**). Western blotting confirmed expression of the Hsh155$^{V502F}$ fusion protein used in the two-hybrid assay (not shown). Other tested LBU-mutants with no or weaker splicing phenotypes (Hsh155$^{D563G, E571K}$) did not significantly alter the two-hybrid interaction.

Recent cryo-EM structures of assembled, B complex spliceosomes also place the tri-snRNP protein Prp3 near Hsh155 and the LBU-mutational site (**Fig 1B**). In these structures, a putative interaction forms between the N-terminal domain of Prp3 and Hsh155 [30]. Prp3 potentially serves as an anchor point for the attachment of the U2 snRNP to the rest of the complex. However, the resolution of this interaction was poor and precluded modeling of Prp3 amino acids

**Fig 6. Yeast two-hybrid assay for interactions between Hsh155, Prp5, and Prp3.** Hsh155[WT] or LBU-mutant proteins were fused to the AD of the two-hybrid reporter and either Prp5 (A) or Prp3 (B) were fused to the reporter BD. +His represents non-selective media containing histidine. -His represents selective media lacking histidine in which growth indicates a two-hybrid interaction. Each two-hybrid assay was repeated three times.

in this region or a molecular description of the Hsh155/Prp3 interface. To our knowledge, an interaction between Hsh155 and Prp3 has not yet been biochemically validated.

We used our two-hybrid assay to test the putative interaction formed between Prp3 and Hsh155. Hsh155 and Prp3 also form a strong two-hybrid interaction (**Fig 6B**). Intriguingly, this interaction is also disrupted by the Hsh155[V502F] allele but not by LBU-mutants located closer to the observed Hsh155/Prp3 interface (Hsh155[D563G, E571K]; **Fig 1A** and **Fig 6B**). Together, this data suggests that V502F mutation may be causing conformational changes in the Hsh155 structure, disrupting the intermolecular interactions with both Prp3 and Prp5 splicing partners as well as the genetic interaction with Prp2. The impact of Hsh155[V502F] on splicing is likely complex since it potentially interferes with complexes forming at different stages in the splicing reaction.

## Discussion

In this work, we tested the impact of the SF3b1 mutations associated with LBU cancers on splicing in yeast. All of the LBU-mutants supported yeast splicing and viability when incorporated into the yeast homolog Hsh155. Given the high conservation of the splicing machinery between yeast and humans, this suggests that these mutations are not loss of function in humans but alter splicing in more subtle ways. In support of this, the Hsh155[V502F] and Hsh155[D563G] mutants decrease $Cu^{2+}$-tolerance of yeast in the presence of ACT1-CUP1 reporter pre-mRNAs harboring non-consensus BS. These results are similar to those previously observed for MDS mutants in Hsh155 HR4-8 and demonstrate that mutations in HR9 and HR11 can also influence BS usage [9, 21].

It has previously been noted that cancer-associated mutations in SF3b1 tend to follow a periodicity of 40 amino acids (approximately the size of one HR) and that the mutations tend

to cluster along an edge of the protein [24]. SF3b1$^{L833F}$/Hsh155$^{V502F}$ deviates from this trend as the amino acid is located within the interior of the protein with the side chain packed at the interface of the two α-helices comprising HR9. Cretu and co-workers speculated that some cancer-associated mutations in SF3b1 may result in structural perturbations of the HEAT repeat superhelix [6]. It is possible that these types of structural changes result in the splicing and protein-interaction phenotypes we observe with the Hsh155$^{V502F}$ mutation. Destabilization of the HR9 fold by substitution of a larger amino acid at the α-helix interface may also explain why this particular mutation exhibited stronger phenotypes than did mutations occurring on the surface of the protein. While it is likely that similar destabilization occurs in human SF3b1 with this same substitution to a larger amino acid, it remains to be seen if SF3b1$^{L833F}$ phenocopies our observations in yeast.

HR domains such as the one found in SF3b1/Hsh155 are widely used to mediate protein-protein interactions [29]. SF3b1/Hsh155 is no exception, and cryo-EM analysis of yeast and human spliceosome structures have revealed a number of splicing factors which interact with the SF3b1/Hsh155 HR domain [11] Many of these interactions are transient and likely occur consecutively in an ordered fashion: the HR binds to and then releases Prp5 during spliceosome assembly [9, 21] the HR then interacts with Prp3 in B complex spliceosomes before activation and release of the U4 snRNP (including Prp3) [30], and finally Prp2 docks onto the HR to release U2 proteins including SF3b1/Hsh155 prior to splicing catalysis [14, 15] The ability of the Hsh155$^{V502F}$ mutation studied here as well as other MDS-mutations in Hsh155 [9, 21] to change yeast two-hybrid and genetic interactions with these splicing factors suggests that SF3b1 mutations can perturb splicing at multiple stages. While most models for SF3b1 dysfunction in cancer have focused on steps involved in BS recognition and U2 loading [19, 20, 22], it is possible that some degree of splicing dysregulation occurs through disruption of later steps. For example, some alternative BS may be used due to their ability to assemble spliceosomes that can be successfully activated through altered interactions between SF3b1 and the human homologs of Prp2 and/or Prp3. Alternatively, other BS may be avoided due to failure to correctly activate spliceosomes rather than possessing a defect in U2 loading *per se*. The reversibility of spliceosome assembly [31] the splicing reaction [32], as well as proofreading and disassembly pathways occurring in later splicing complexes [33–35] support the notion that some degree of splicing regulation may occur at later steps. Indeed, evidence for late-stage regulation of splicing has been reported for the human splicing machinery [36, 37]. Understanding how mutant alleles of SF3b1 give rise to particular RNA isoforms and lead to disease may therefore require detailed investigations into the entire splicing pathway from start to finish.

## Methods and materials

*Saccharomyces cerevisiae* strains used in this work were derived from 46α and BJ2168 parental lines. The details of the strains and plasmids used are listed in **S1** and **S2 Tables**. Yeast transformations, plasmid shuffling, and growth were carried out using standard techniques and media [38].

### Site-directed mutagenesis

LBU-mutant Hsh155 genes were generated using an inverse polymerase chain reaction (PCR) with Phusion DNA polymerase (New England Biolabs; Ipswich, MA, USA). The PCR was performed for 16 cycles, and PCR products were treated with DpnI (New England Biolabs; Ipswich, MA) to remove the template. The PCR products were then 5' phosphorylated and ligated using T4 polynucleotide kinase (New England Biolabs; Ipswich, MA, USA) and T4 DNA ligase

(New England Biolabs; Ipswich, MA, USA) respectively. The ligated products were transformed into Top10 competent cells. Individual colonies were screened by sequencing to identify the desired mutation.

## Temperature growth assays

Yeast strains expressing WT or mutant proteins were grown to mid-log phase at 30˚C in YPD liquid media. Cell growth in liquid medium was quantified by measuring the optical density at 600 nm. Equal volumes of the cells at $OD_{600} = 0.5$ were stamped onto YPD agar plates and incubated at 23˚C, 30˚C or 37˚C for three days or at 16˚C for ten days.

## ACT1-CUP1 assays

Copper resistant growth assays were performed using yeast strains expressing WT Hsh155 or mutant Hsh155 alleles and ACT1-CUP1 reporters [26]. Cells were grown to mid-log phase at 30˚C in an appropriate media, adjusted to $OD_{600} = 0.5$, and equal volumes were spotted onto 30 plates with $CuSO_4$ concentrations of 0, 0.025, 0.05, 0.075, 0.1, 0.15, 0.2, 0.25, 0.3, 0.4, 0.5, 0.6, 0.7, 0.8, 0.9, 1.0, 1.1, 1.2, 1.3, 1.4, 1.5, 1.6, 1.7, 1.8, 1.9, 2.0, 2.25 or 2.5 mM. Plates were incubated at 30˚C for three days before imaging.

## Primer extension

Cells were grown at 30˚C in 10 mL yeast synthetic dropout liquid media until $OD_{600}$ reached 0.5–0.8, and 10 $OD_{600}$ units were harvested by centrifugation. RNA was isolated using a MasterPure Yeast RNA Purification Kit (Epicentre BioTechnologies; Madison, WI, USA). Primer extensions reactions were performed using the primers YAC6 (`5'/IRD700/GGCACTCAT GACCTTC-3'`) and YU6 (`5'/IRD700/GAACTGCTGATCATCTCTG-3'`) (IDT DNA, Coralville, IA, USA)[9, 39]. Primer extension reactions contained 10 µg total cellular RNA, 10 U Superscript III (ThermoFisher Scientific; Waltham, MA, USA), 100 nM IR dye-labeled YAC6 primer, and 20 nM IR labeled YU6 primer. Assembled extension reactions were incubated at 55˚C for 1h, and extension products were analyzed using denaturing PAGE (7% acrylamide: bisacrylamide (19:1), 8M urea, 1× TBE). Gels were imaged using an Amersham Typhoon NIR biomolecular imager, and band intensities were quantified using ImageQuant software (GE Healthcare Life Sciences; Chicago, IL, USA).

## Yeast two-hybrid assays

LBU-mutants of Hsh155 (V502F, D563G, and E571K) were derived from a plasmid (pAAH0499), which generates a GAL4 activation domain fused with Hsh155 [9] The open reading frames of Prp5 and Prp3 were fused to the C-terminus of the GAL4-DNA binding domain in plasmid pGBKT7 [9] Each pair of the plasmids was transformed into the *S. cerevisiae* strain Y2H GOLD (Takara Biosciences; Mountain View, CA). The interactions between Prp5 and Hsh155 mutants and Prp3 and Hsh155 mutants were examined by growth on selective media plates. Briefly, yeast strains expressing both fusions were grown to mid-log phase at 30˚C in liquid media lacking leucine and tryptophan to maintain selection for the plasmids. Cell growth in liquid medium was quantified by measuring the optical density at 600 nm, and the $OD_{600}$ was adjusted to 0.5. Equal volumes of the cells were stamped onto media lacking histidine, leucine, and tryptophan and incubated at 30˚C for three days before imaging.

## Supporting information

**S1 Raw images. Uncropped gel images used to generate Fig 3.**
(PDF)

**S1 Table. Yeast strains used in this study.**
(PDF)

**S2 Table. Plasmid used in this study.**
(PDF)

## Author Contributions

**Conceptualization:** Harpreet Kaur, Aaron A. Hoskins.

**Data curation:** Harpreet Kaur, Joshua C. Paulson, Sarah McMillan.

**Formal analysis:** Harpreet Kaur, Brent Groubert, Sarah McMillan.

**Funding acquisition:** Aaron A. Hoskins.

**Investigation:** Harpreet Kaur, Brent Groubert, Joshua C. Paulson.

**Resources:** Harpreet Kaur, Joshua C. Paulson.

**Supervision:** Aaron A. Hoskins.

**Visualization:** Harpreet Kaur, Brent Groubert, Sarah McMillan.

**Writing – original draft:** Harpreet Kaur.

**Writing – review & editing:** Harpreet Kaur, Aaron A. Hoskins.

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
