## [Decision Letter · Decision Letter 0]

26 Feb 2020

PONE-D-20-02992

Impact of Cancer-Associated Mutations in Hsh155/SF3b1 HEAT Repeats 9-12 on pre-mRNA Splicing in Saccharomyces cerevisiae

PLOS ONE

Dear Dr hoskins,

Thank you for submitting your manuscript to PLOS ONE. Manuscript was reviewed by two experts who raised minor but addressable concerns. Therefore, we invite you to submit a revised version of the manuscript that addresses the points raised during the review process.

ACADEMIC EDITOR: 

Please elaborate your discussion to address the raised points in as much detail as possible. 

We would appreciate receiving your revised manuscript by Apr 11 2020 11:59PM. To enhance the reproducibility of your results, we recommend that if applicable you deposit your laboratory protocols in protocols.io, where a protocol can be assigned its own identifier (DOI) such that it can be cited independently in the future. For instructions see: http://journals.plos.org/plosone/s/submission-guidelines#loc-laboratory-protocols

We look forward to receiving your revised manuscript.

Kind regards,

Ravindra N Singh, Ph.D.

Academic Editor

PLOS ONE

Journal Requirements:

Reviewers' comments:

Reviewer's Responses to Questions

**Comments to the Author**

1. Is the manuscript technically sound, and do the data support the conclusions?

Reviewer #1: Yes

Reviewer #2: Yes

2. Has the statistical analysis been performed appropriately and rigorously? 

Reviewer #1: N/A

Reviewer #2: Yes

3. Have the authors made all data underlying the findings in their manuscript fully available?

Reviewer #1: Yes

Reviewer #2: Yes

4. Is the manuscript presented in an intelligible fashion and written in standard English?

Reviewer #1: Yes

Reviewer #2: Yes

5. Review Comments to the Author

Reviewer #1: Aaron Hoskins and co-authors use yeast genetic systems to investigate the functional consequences of a subset of cancer-associated mutations in the pre-mRNA splicing factor SF3B1 (called Hsh155 in yeast). They focus on a subset of mutations concentrated in HR9-12 and associated with AML, bladder and uterine cancers. Incorporation of the mutants in the yeast homologue has no detectable effect on cell proliferation or temperature sensitivity. Likewise, most mutations had little effect on splicing of an ACT1-CUP1 reporter. Two mutations had greater effects. In particular, Hsh155V502F was defective in splicing ACT1-CUP1 when the BPS was degenerate. Hsh155D563G also showed a slight defect. The authors focus on Hsh155V502F for additional studies. The effects of Hsh155V502F were additive with other, previously-characterized MDS-associated alleles. Hsh155V502F reduced genetic/two-hybrid interactions with Prp2 and Prp5 helicases or Prp3 U4/U6 subunit compared to WT Hsh155. These results provide new insights to the field, since few studies of mutations in SF3B1 HR9-12 are available compared to the well-characterized, MDS-associated mutations of HR4-8. The audience is broad because SF3B1 is important for gene expression and pre-mRNA splicing and is frequently dysregulated in human disease.

Points to consider:

1) Please include and discuss the frequency of the mutations studied in various diseases, particularly the V502F mutation to the Introduction/Discussion and Figure 1 or a new Table.

2) Please include how the alignment in Figure 1 was achieved to the figure legend or Methods.

3) I don’t believe that it has yet been shown whether SF3B14 and U2AF65 cooperate or compete using full length proteins and/or in the context of the holo-SF3B1/RNA. Work by the Sattler/Luhrmann groups (RNA 2006) shows SF3B14 and U2AF65 UHM domain can bind simultaneously to SF3B1, but do not bind cooperatively. Please rephrase the statement on Pg. 4, line 64 of Introduction.

4) The Results would be strengthened if a change in splicing of endogenous genes was shown in addition to the ACT1-CUP1 reporter.

5) Only one of the mutations studied, V502F, has a significant effect on splicing. Please discuss why this would be. Is it reasonable to anticipate that the mutations have different functional and magnitude of effects?

6) The mutations are shown mapped on the HSH155 structure, but the structure has not been related to the consequences of the mutations in the text. Please discuss possible structural rationales for distinctive V502F effects and cite precedents. For example, the Cretu SF3B1 structure Mol Cell (2016) discuss the likely unfolding or partial unfolding of HEAT repeats by the MDS-associated SF3B1 mutations. It is hard to see in the figures, but it appears that V502 is partially buried between HEAT repeats, such that a mutation to bulky F could locally unfold this region. Unfolding could explain the compromised interactions with different Prp2 and Prp3 subunits as well as sensitivity to the BPS sequence.

7) Since V502 is Leucine in humans, would the authors expect the effects of an L502F mutation to be as severe as in yeast?

Reviewer #2: In the manuscript “Impact of Cancer-Associated Mutations in Hsh155/SF3b1 HEAT Repeats 9-12 on Pre-mRNA Splicing in Saccharomyces cerevisiae,” Hoskins and colleagues analyze the effect of incorporating several cancer-relevant mutations into the yeast homolog of SF3b1, Hsh155. The authors find that the hotspot mutation V502F affects splicing of introns with non-consensus branch site (BS) sequences and shows genetic interactions and altered physical interactions with several splicing factors.

The data are clearly presented and nicely demonstrate complex interactions between Hsh155 and multiple splicing factors that act at different stages in assembly. There are two questions that are raised by the data that the authors should address:

1. Figure 3B. What is the prominent band in the middle of the primer extension gel in the mutant lanes? Could it be lariat intron, where primer extension stops at a branch or evidence of use of some alternative BS?

2. Figure 2C. It is interesting that of the mutations tested, mutation of the BP adenosine (A259G) has the least sensitivity to copper when compared to the other mutations? The authors should address this, perhaps in the discussion.

While obviously beyond the scope of this paper, it would be interesting to know if splicing of endogenous ICGs with non-consensus branch sites is affected in this mutant and if alternative BS usage can be detected.

6. PLOS authors have the option to publish the peer review history of their article (what does this mean?). If published, this will include your full peer review and any attached files.

Reviewer #1: No

Reviewer #2: No

---

## [Author Response · Author response to Decision Letter 0]

19 Mar 2020

Reviewer #1: Aaron Hoskins and co-authors use yeast genetic systems to investigate the functional consequences of a subset of cancer-associated mutations in the pre-mRNA splicing factor SF3B1 (called Hsh155 in yeast). They focus on a subset of mutations concentrated in HR9-12 and associated with AML, bladder and uterine cancers. Incorporation of the mutants in the yeast homologue has no detectable effect on cell proliferation or temperature sensitivity. Likewise, most mutations had little effect on splicing of an ACT1-CUP1 reporter. Two mutations had greater effects. In particular, Hsh155V502F was defective in splicing ACT1-CUP1 when the BPS was degenerate. Hsh155D563G also showed a slight defect. The authors focus on Hsh155V502F for additional studies. The effects of Hsh155V502F were additive with other, previously-characterized MDS-associated alleles. Hsh155V502F reduced genetic/two-hybrid interactions with Prp2 and Prp5 helicases or Prp3 U4/U6 subunit compared to WT Hsh155. These results provide new insights to the field, since few studies of mutations in SF3B1 HR9-12 are available compared to the well-characterized, MDS-associated mutations of HR4-8. The audience is broad because SF3B1 is important for gene expression and pre-mRNA splicing and is frequently dysregulated in human disease.

We thank the reviewer and agree that this work provides new insights and is of interest to a broad audience. 

Points to consider:

1) Please include and discuss the frequency of the mutations studied in various diseases, particularly the V502F mutation to the Introduction/Discussion and Figure 1 or a new Table.

We have analyzed the data by Seiler et al. with respect to the diseases mentioned in the introduction and have included the number of mutations observed in HR9-12 along with the number of samples studied and more detailed information on SF3b1L833F/Hsh155V502F (pg 5, Lines 85-90). We don’t wish to delve too deeply into allele frequency since this is not our area of expertise and the data of Seiler et al. were limited to those contained in The Cancer Genome Atlas.

2) Please include how the alignment in Figure 1 was achieved to the figure legend or Methods.

We have added this information to the figure legend.

3) I don’t believe that it has yet been shown whether SF3B14 and U2AF65 cooperate or compete using full length proteins and/or in the context of the holo-SF3B1/RNA. Work by the Sattler/Luhrmann groups (RNA 2006) shows SF3B14 and U2AF65 UHM domain can bind simultaneously to SF3B1, but do not bind cooperatively. Please rephrase the statement on Pg. 4, line 64 of Introduction.

We apologize for the confusion. This sentence has been rephrased. 

4) The Results would be strengthened if a change in splicing of endogenous genes was shown in addition to the ACT1-CUP1 reporter.

This is beyond the scope of the current manuscript.

5) Only one of the mutations studied, V502F, has a significant effect on splicing. Please discuss why this would be. Is it reasonable to anticipate that the mutations have different functional and magnitude of effects?

See below.

6) The mutations are shown mapped on the HSH155 structure, but the structure has not been related to the consequences of the mutations in the text. Please discuss possible structural rationales for distinctive V502F effects and cite precedents. For example, the Cretu SF3B1 structure Mol Cell (2016) discuss the likely unfolding or partial unfolding of HEAT repeats by the MDS-associated SF3B1 mutations. It is hard to see in the figures, but it appears that V502 is partially buried between HEAT repeats, such that a mutation to bulky F could locally unfold this region. Unfolding could explain the compromised interactions with different Prp2 and Prp3 subunits as well as sensitivity to the BPS sequence.

See below.

7) Since V502 is Leucine in humans, would the authors expect the effects of an L502F mutation to be as severe as in yeast?

Reviewer comments 5-7 all address interpretation of the V502F mutation. We have added a short paragraph to address these points to the discussion (pg 12, lines 249-262). In summary, this amino acid is located at the interface between the two alpha helices that comprise HR9. We predict that the packing of the alpha helices is disrupted by this mutation (in agreement with hypotheses made by Cretu) and it is likely this disruption also occurs with humans. Whether or not the human protein phenocopies our observations in yeast is not yet known. 

Reviewer #2: In the manuscript “Impact of Cancer-Associated Mutations in Hsh155/SF3b1 HEAT Repeats 9-12 on Pre-mRNA Splicing in Saccharomyces cerevisiae,” Hoskins and colleagues analyze the effect of incorporating several cancer-relevant mutations into the yeast homolog of SF3b1, Hsh155. The authors find that the hotspot mutation V502F affects splicing of introns with non-consensus branch site (BS) sequences and shows genetic interactions and altered physical interactions with several splicing factors.

The data are clearly presented and nicely demonstrate complex interactions between Hsh155 and multiple splicing factors that act at different stages in assembly. There are two questions that are raised by the data that the authors should address:

We thank the reviewer for the kind comments on our work.

1. Figure 3B. What is the prominent band in the middle of the primer extension gel in the mutant lanes? Could it be lariat intron, where primer extension stops at a branch or evidence of use of some alternative BS?

We have not identified this band. It is likely to be lariat intron-3’ exon based on its size; however, we do not wish to speculate at this stage. We do not feel it is likely to originate from use of a cryptic BS since any other potential BS in the ACT1-CUP1 reporter would be extremely weak—although we have not formally excluded this possibility. 

Importantly, regardless of its identity our interpretation of the data remain valid: the Hsh155V502F and Hsh155D563G mutants produce less of the correct mRNA product than WT and this is consistent with ACT1-CUP1 growth data. We have added an asterisk to the figure and have included clarifying language to the figure legend. 

2. Figure 2C. It is interesting that of the mutations tested, mutation of the BP adenosine (A259G) has the least sensitivity to copper when compared to the other mutations? The authors should address this, perhaps in the discussion.

We have previously observed this phenomenon with other SF3b1/Hsh155 mutations (see studies by Carrocci and Hoskins). The most likely explanation is that since A259G would be excluded from the U2/BS duplex (i.e., “flipped out”) that position is not specifically recognized by the protein. This work is consistent with in-depth studies by Query and coworkers on BS recognition as well as cryo-EM structures of the protein bound to U2/BS duplexes. We have added a short discussion of this to page 7, lines 141-146. 

While obviously beyond the scope of this paper, it would be interesting to know if splicing of endogenous ICGs with non-consensus branch sites is affected in this mutant and if alternative BS usage can be detected.

We agree that this is beyond the scope of the current manuscript.

---

## [Editor Report · Decision Letter 1]

23 Mar 2020

Impact of Cancer-Associated Mutations in Hsh155/SF3b1 HEAT Repeats 9-12 on pre-mRNA Splicing in Saccharomyces cerevisiae

PONE-D-20-02992R1

Dear Dr. hoskins,

We are pleased to inform you that your manuscript has been judged scientifically suitable for publication and will be formally accepted for publication once it complies with all outstanding technical requirements.

With kind regards,

Ravindra N Singh, Ph.D.

Academic Editor

PLOS ONE
---

## [Editor Report · Acceptance letter]

24 Mar 2020

PONE-D-20-02992R1 

Impact of Cancer-Associated Mutations in Hsh155/SF3b1 HEAT Repeats 9-12 on pre-mRNA Splicing in *Saccharomyces cerevisiae*

Dear Dr. Hoskins:

I am pleased to inform you that your manuscript has been deemed suitable for publication in PLOS ONE. Congratulations! Your manuscript is now with our production department. 

With kind regards,

on behalf of

Dr. Ravindra N Singh 

Academic Editor

PLOS ONE